# Effective Electrical Properties and Fault Diagnosis of Insulating Oil Using the 2D Cell Method and NSGA-II Genetic Algorithm

**DOI:** 10.3390/s23031685

**Published:** 2023-02-03

**Authors:** José Miguel Monzón-Verona, Pablo González-Domínguez, Santiago García-Alonso

**Affiliations:** 1Electrical Engineering Department (DIE), University of Las Palmas de Gran Canaria, 35017 Las Palmas de Gran Canaria, Spain; 2Institute for Applied Microelectronics, University of Las Palmas de Gran Canaria, 35017 Las Palmas de Gran Canaria, Spain; 3Department of Electronic Engineering and Automatics (DIEA), University of Las Palmas de Gran Canaria, 35017 Las Palmas de Gran Canaria, Spain

**Keywords:** fault diagnosis, electrical insulating oil, effective electrical properties, wireless sensors, genetic algorithms, cell method

## Abstract

In this paper, an experimental analysis of the quality of electrical insulating oils is performed using a combination of dielectric loss and capacitance measurement tests. The transformer oil corresponds to a fresh oil sample. The paper follows the ASTM D 924-15 standard (standard test method for dissipation factor and relative permittivity of electrical insulating liquids). Effective electrical parameters, including the *tan δ* of the oil, were obtained in this non-destructive test. Subsequently, a numerical method is proposed to accurately determine the effective electrical resistivity, *σ*, and effective electrical permittivity, *ε*, of an insulating mineral oil from the data obtained in the experimental analysis. These two parameters are not obtained in the ASTM standard. We used the cell method and the multi-objective non-dominated sorting in genetic algorithm II (NSGA-II) for this purpose. In this paper, a new numerical tool to accurately obtain the effective electrical parameters of transformer insulating oils is therefore provided for fault detection and diagnosis. The results show improved accuracy compared to the existing analytical equations. In addition, as the experimental data are collected in a high-voltage domain, wireless sensors are used to measure, transmit, and monitor the electrical and thermal quantities.

## 1. Introduction

The increasing demand for electric power in the world—electricity consumption increased by 129.6% between 1990 and 2021 [1]—brings with it a greater need for reliable power equipment, such as transformers or on-load tap changers, among many others. In addition, as a result of the rapid increase in power transmission voltage, the performance requirements for power transformer insulation are also increasing [2].

Power transformers constitute the highest equipment cost in electrical substations, accounting for almost 60% of the total investment [3]. The threshold temperature for transformer operation is 80 °C. Above this temperature, transformer lifetime is halved for each temperature increase between 6 and 7 °C [4].

In transformers, high amounts of dielectric liquids are used for phase-to-phase/phase-to-ground insulation practices, as well as cooling liquid to evacuate heat generated due to hysteresis and eddy current losses in iron, as well as the losses due to the Joule effect in the transformer coils. Various insulating liquids with different chemical, physical, and dielectric properties are used in electrical installations to provide continuous operations. However, it is well known that some of the electrical faults occurring in power equipment arise from the deterioration of these dielectric liquids [5].

The regular and trouble-free operation of these components is vital to ensure a continuous energy supply and economic profitability of the power system. A power outage resulting from transformer failure, which is one of the costliest and most important parts of an electric power system, could lead to production interruption and considerable economic losses.

It is also of the utmost importance to ensure an adequate level of environmental safety, given that during their operation the stresses that high-voltage devices and, consequently, insulating materials operate under can be critical [6].

A set of techniques for the monitoring and diagnosis of faults that affect the life cycle of these important elements is required. These techniques include the following: dissolved gas analysis, oil quality testing, infrared thermography testing, power factor test, tan delta testing, and insulating oil breakdown testing, among others.

The optimization algorithm that is developed in this work allows for the determination of effective electrical permittivity and conductivity properties, *ε* and *σ*, on the basis of the global data obtained in the tan delta test and a distributed parameter model.

The analytical equations presented in [5] are only an approximation, since they do not take into account the geometry of the test device container (TDC) and assume, beforehand, a known value of the electrical permittivity of the TDC without oil.

The knowledge of these dielectric properties allows for the planning of preventive maintenance to avoid future transformer breakdowns in power systems [2], and can be used to perform a finite element numerical analysis to determine the electrical field distribution results for AC breakdown strength testing [7]. Activation energy, *E_ac_*, [8], is an indicator that can be used to determine properties, such as electrical conductivity, as a function of temperature. In [9], the *E_ac_* profiles of a nanostructured alumina polycarbonate composite for the improvement of electrical insulation conditions in materials were evaluated. The insulation system in a transformer degrades over time and needs periodic monitoring for the uninterrupted operation of a power system network. A non-intrusive and non-destructive testing method using an S-band pyramidal horn antenna is introduced in [10].

The timely identification of the aging stage of the transformer can effectively ensure its safe operation and prevent aging faults [11].

In this paper, a non-destructive testing (NDT) analysis of the quality of electrical insulating oils is performed using a combination of dielectric loss and capacitance measurement tests. The transformer oil corresponds to a fresh oil sample. The article follows the ASTM D 924-15 [12] standard (standard test method for dissipation factor and relative permittivity of electrical insulating liquids). The IEC 60,247 standard [13] is equivalent to ASTM D 924-15. Effective electrical parameters, such as the tan δ of the oil, are obtained.

Subsequently, a numerical method is proposed to accurately determine the effective electrical resistivity, *σ*, and effective electrical permittivity, *ε*, of an insulating mineral oil from the data obtained in the experimental analysis. The cell method (CM) and non-dominated sorting in genetic algorithm II (NSGA-II), a multi-objective genetic algorithm, are used. Therefore, in this paper, a new numerical tool to accurately obtain the effective electrical parameters of transformer insulating oils is provided for fault detection and diagnosis. The distributed parameter model and GA-based algorithm are used with the aim of improving accuracy compared to existing analytical equations that are used. As the experimental data are collected in a high-voltage domain, wireless sensors are used to measure, transmit, and monitor the electrical and thermal magnitudes.

The optimization algorithm used in this paper to determine the effective electrical parameters can be classified as a metaheuristic algorithm. Such algorithms are used to solve complex problems in various fields of electrical engineering. Intensification and diversification are their key elements. Most of these metaheuristic algorithms are inspired by the process of biological evolution. Of these, the genetic algorithm (GA) is the best known [14].

The multi-objective GA (MOGA) is a modified version of the simple GA. The MOGA differs from the GA in terms of the objective function assignment, but its other steps are similar to those of the GA. The main feature of the MOGA is to generate the Pareto optimal front in the objective space such that one objective function is not improved by worsening the other objective functions [14]. The concept of Pareto dominance was introduced in the MOGA in [15], the paper in which the first MOGA was developed.

Proposed in [16], the NSGA-II is an improved version of the NSGA and is a widely used algorithm. It uses a selection scheme in which the population of parents is compared with the population of offspring. The NSGA-II, in addition to the use of elitism, is more computationally efficient than the NSGA. It is a highly competitive algorithm in convergence to the Pareto optimum.

GAs have been used previously to detect incipient transformer oil faults [17], while GA-based predictive models have been used as an auxiliary indicator method to determine the aging condition of transformer polymer insulation [18]. In [19], a GA was used for accurate measurements of partial discharge. The use of GAs in design is well known. In [20], a GA was used with the finite element method for the design of transformers. In [21], a new multi-objective optimization tool was provided for the design of low-power-pressure microsensors using NSGA-II.

However, in the literature, to the authors’ knowledge, the two-dimensional cell method (2D-CM) has not been used together with the NSGA-II and ASTM D 924-15 to determine accurate measurements for the ε and σ of an insulating mineral oil.

This paper is divided into the following sections. In Section 2, “Theoretical background”, the TDC is described, and the distributed and lumped parameter models are developed. In Section 3, “Non-destructive insulation tests”, the electrical connection scheme is presented, the Schering bridge used for the experimental measurements is described, the high-voltage tests are presented, and the parameters used in the following section are obtained. In Section 4, a description is provided of the fit of the effective oil parameters by means of the 2D-CM and NSGA-II. Finally, in Section 5, “Conclusions”, the main conclusions obtained are presented.

## 2. Theoretical Background

A standard TDC, described in detail in Section 2.1, was used to conduct the experimental analysis of the quality of electrical insulating oils.

The effective electrical properties of mineral oil (σ and ε) were obtained based on two types of models. The first was the distributed parameter model of the TDC, which is developed in Section 2.2. The finite formulation (FF) of the electromagnetic equations [22,23] was used, along with the numerical CM to solve these equations [24,25].

The second model was the TDC lumped parameter model, developed in Section 2.3. This model was necessary to be able to relate the experimental measurements performed in the high-voltage laboratory (global magnitudes of electric potential difference and intensity of electric current) with the magnitudes of the distributed parameter model used by the CM (distributed electric potentials throughout the TDC domain).

### 2.1. Test Device Container

The TDC is composed of stainless steel and conforms to the ASTM D 924-15 standard. It is shaped similar to a cylindrical capacitor with an external and internal cylindrical plate. The external plate constitutes the container in which the oil is housed. The inner plate is submerged in the oil and rests on the external plate through a circular transparent methacrylate lid. The inner plate is connected to a high voltage through an upper handle. Figure 1 shows a schematic diagram of the two elements that comprise the TDC.

In summary, the two main elements of the TDC are the oil container, which, during the tests, is connected to the ground, and an inner cylindrical plate, which is connected to a high voltage, with its methacrylate cover and ring.

The TDC for liquid insulants (max. 10 kV) that was used was a Haefely Hipotronics 6835 test cell [24,26]. It allows tests on the dielectric properties of liquid insulants and can be used to determine the dissipation factor (*tan δ*). It is particularly suitable for on-site maintenance measurements on the insulating oil of electrical apparatus, including transformers and oil circuit-breakers. Its main properties are presented in Table 1.

### 2.2. Distributed Parameter Model

For transformer oils, the electrical constitutive equation is a complex equation based on the Fower–Nordheim theory [27]. In most dielectric materials, the free-carrier conduction current is relatively low since their conductivity is usually several orders of magnitude below that of a metal or semiconductor. In new transformer oils, at 50 °C it is usually around 1 × 10^−13^ S/m and in used oils in the order of 1 × 10^−11^ S/m. This is because oil conductivity mainly depends on the conductive particles inside it [5].

The oil electrical permittivity is *ε* = *ε_r_* × 8.854187818 × 10^−12^ F/m, with the oil relative permittivity, *ε_r_*, being between 2 and 4 [5]. A higher relative permittivity value for an insulating liquid indicates that it is less exposed to electric field stresses. This condition is an advantage for the quality of the insulation. In addition, the higher permittivity of mineral oil is better at building a more uniform electric field.

In this paper, a conductive model was considered, in which the volumetric current density $\vec{J}$ was directly proportional to the electric field $\vec{E}$ [27].

Using the FF and CM as the numerical methods, the equation to solve the quasi-electrostatic problem is the following:(1)$$\tilde{D}M_{\sigma }G\varphi +\frac{\partial }{\partial t}\tilde{D}M_{\varepsilon }G\varphi =0$$
where $\tilde{D}$ is the volumes–faces incidence matrix in the dual mesh. This matrix represents the discrete divergence associated with the dual volume. $\tilde{D}=-G^{t}$ is met with matrix $M_{\sigma }$ being the electrical constitutive matrix. Matrix $M_{\varepsilon }$ is the constitutive matrix of electrical permittivity and G is the edges–nodes incidence matrix in the primal mesh. In the primal mesh, the electrical potentials of the nodes are in the vector φ.

If the electrodes work at the frequency *f* = 50 Hz (with the angular frequency *ω* = *2πf*), then Equation (1) is reduced to Equation (2) in the frequency domain. In this equation, *σ* and *ε* appear simultaneously. This is the equation that must be programmed together with the global electrode complex current vector ${\bar{I}}_{t}$, which is calculated by Equation (3), where $I_{c}$ is an incidence vector of the relative cut between the edges of the mesh of the oil volume and the surface of one of the TDC cylindrical plates [25,28]. The sum of all the currents in that cut is equal to the total current that enters or leaves each of the TDC electrodes.
(2)$$\tilde{D}M_{\sigma }G\bar{\varphi }+j\omega \tilde{D}M_{\varepsilon }G\bar{\varphi }=0$$
(3)$${\bar{I}}_{t}=-I_{c\;}\left(M_{\sigma }G+j\omega M_{\varepsilon }G\right)\bar{\varphi }$$

The unknowns are all the complex electrical potentials $\bar{\varphi }$ associated with the nodes of the primal mesh.

In addition to the optimization algorithm, which is presented in Section 4, the global magnitudes are needed to determine the effective parameters σ and ε. They are obtained in the post-processing stage, once Equation (2) has been solved.

The first of these is the current, ${\bar{I}}_{t}$, associated with the surface of the electrodes, as described in Equation (3).

Associated with the entire domain, the second global magnitude is the average electrical energy stored over time (rms value) in the oil, $W_{e}$, according to the following equation [29]:(4)$$W_{e}=\int_{vol}\frac{1}{4}\bar{E}\varepsilon {\bar{E}}^{*}dv$$
where $\bar{E}$ and ${\bar{E}}^{*}$ are, respectively, the complex electric field and its conjugate in the whole domain.

The third global magnitude corresponds to the total losses due to the Joule effect averaged over time (rms value) in the oil, $P_{Joule}$. They are given by:(5)$$P_{Joule}=\int_{vol}\frac{1}{2}\bar{E}\sigma {\bar{E}}^{*}dv$$
where σ and ε are analytically related [5] through the following equation:(6)$$tan\;\delta =\frac{\sigma }{\omega \varepsilon }$$
where *tan δ* is the loss factor.

Figure 2a shows the external cylindrical plate and Figure 2b the mesh generated with Gmsh software [30]. In this region, all the nodes of the conducting region were set at zero potential as the boundary condition.

Figure 3a shows the inner cylindrical plate which, in the numerical simulations, is connected to a high voltage. This electric potential corresponds to each of the electric potentials used in the experiments conducted in the laboratory. The boundary condition corresponding to this region is to set all the nodes of the conducting region at a high voltage.

All equations were programmed in C++. Equation (2) is symmetric. As the matrices are sparse and large, numerical methods based on the Krylov subspaces were used. These algorithms were implemented with the PETSc numerical package [31]. This numerical package uses parallel processing, which reduces the calculation times. In particular, the linear solver used was the generalized minimal residual (GMRES) algorithm.

The main characteristics of the computer used in the simulations were as follows: computer model = X399 AORUS PRO; architecture = x86_64; total memory = 128 GB; processors = 24; CPU = 2185.498 MHz; threads per core = 2; and cores per socket = 12.

To verify the CM model expressed with Equation (2), a simulation was performed with the TDC without oil (Figure 4). The result obtained was compared with the value provided by the manufacturer and was observed to coincide.

The intensity of the electric field was the negative gradient of the electric potential, which was obtained by solving Equation (2). The intensity of the electric field was used to calculate the stored energy and heat losses, as observed in Equations (4) and (5), which were necessary in the calculation of the effective parameters, σ and ε, as explained in Section 4.1.

Figure 4b shows the distribution of the electric field intensity module corresponding to one of the optimization experiments conducted (AG3 in Section 4.1). The values of electrical conductivity and relative permittivity of this distribution were 20.7 pS/m and 2.11, respectively.

To reduce the simulation time in the optimization process, the axial symmetry of the problem was used. This benefitted the calculation process due to the high number of simulations conducted in the process of searching for effective properties. For this simulation, Figure 5a shows the distribution of nodes in the TDC. The number of triangles is 27,842. The number of nodes is only 14,165, in a cross-section, compared to the 20,269 nodes of the 3D mesh in the whole volume. The size of the tetrahedra is much larger than the size of the triangles (see Figure 5b).

With 2D meshing and axial symmetry, a much higher meshing density is achieved. Therefore, the results are more precise with the same computational cost.

The capacity of TDC without oil, according to the model of Equation (2) solved by 2D-CM, produces a value of 108.9 pF, which coincides with the value provided by the manufacturer [24].

As we obtained the capacity of the TDC using CM and the value obtained was equal to that provided by the manufacturer, we validated the CM numerical model that uses geometry and physical laws that govern the problem. Therefore, we could apply any dielectric and any TDC geometry. The proposed method is a general method. It is not a specific method for a particular TDC.

### 2.3. Lumped Parameter Model

In an ideal capacitor, the resistance of the dielectric is infinitely high. This means that, when an alternating current voltage is applied, the electrical current is exactly a 90 degree phase shift with the voltage.

Each insulator showed minor loss values under direct current conditions, with a power $P_{Joule}=\frac{U_{Test}^{2}}{R_{p}},$ where $U_{Test=}\left|{\bar{U}}_{Test}\right|$ is the module of the voltage applied to the TDC and $R_{p}$ is the resistance of the parallel equivalent circuit (see Figure 6b). Under AC conditions, behavior known as dielectric hysteresis loss occurs. Dielectric hysteresis loss is analogous to the magnetic hysteresis loss in iron. In this case, both magnitudes have a phase shift different to 90 degrees. 

The method used to estimate dielectric hysteresis loss in this work was an effective lumped parameter method or circuit model. This model is related to the distributed parameter method observed in Section 2.2 in which σ and ε are involved. In the lumped parameter model, tan δ is calculated. This model includes a calculation of the capacitor $C_{p}$, and the effective resistance of total losses $R_{p}$ (conductive and electrical hysteresis) (see Figure 6a,b).

Its complex admittance takes the following value:(7)$${\bar{Y}}_{p}=G_{p}+j\omega C_{p}$$
where the equivalent conductance $G_{p}=1/R_{p}$, $C_{p}$ is the equivalent capacity of the model and *j* the imaginary unit. In this way, it can be written as:(8)$${\bar{I}}_{t}={\bar{Y}}_{p}{\bar{U}}_{Test}={\bar{I}}_{Rp}+{\bar{I}}_{Cp}=G_{p}{\bar{U}}_{Test}+j\omega C_{p}{\bar{U}}_{Test}$$

Therefore, to identify ${\bar{Y}}_{p}$, it is necessary to know ${\bar{U}}_{Test}$ (which is known because it is the applied voltage). The current ${\bar{I}}_{t}$ is calculated using Equation (3).

With this value, it is possible to estimate the value of the effective tan δ, which is as follows:(9)$$tan\;\delta =\frac{I_{Rp}}{I_{Cp}}=\frac{\sigma }{\omega \varepsilon }$$

The tan δ of an insulating material is expressed as the sum of conduction ($tan\;\delta_{C}$), dielectric hysteresis ($tan\;\delta_{H}$), polarization ($tan\;\delta_{P}$), and ionization ($tan\;\delta_{I}$), as stated in [5]:(10)$$tan\;\delta =tan\;\delta_{C}+tan\;\delta_{H}+tan\;\delta_{P}+tan\;\delta_{I}$$

ASTM D924-15 [10] states that, in general, tanδ values of insulation oils in a good condition are below 0.005. Some of the tanδ values obtained from the scientific literature are 0.001 [32], 0.053 [33], and 0.015 [34]. Higher values are found for aged oils.

As seen in Equation (9), there is a direct relationship between *tan δ* and *σ*, and an inverse relationship with *ε*. This is an approximate analytical equation.

The relationship between the lumped parameters described in this section and the distributed parameters described in Section 2.2 is expressed by the following equations:(11)$$P_{Joule}=\frac{U_{Test}^{2}}{R_{p}}$$
(12)$$W_{e}=\frac{1}{2}U_{Test}^{2}C_{p}$$
where $P_{Joule}$ is the average total thermal loss in the oil over time due to the Joule effect (see Equation (5)), $\mathrm{and}\;W_{e}$ is the average stored electrical energy over time (rms value) in the oil (see Equation (4)). These two equations, (11) and (12), are used later in the fit of the effective parameters σ and ε with the GA program.

## 3. Non-Destructive Insulation Tests

In this section, the non-destructive test ASTM D924-15 is applied. This test establishes both thermal and electrical conditions for the measurement of these quantities [12]. This test method describes the test applied to a fresh electrical insulating oil as well as insulating oil in the service of transformers.

In this test, it must be borne in mind that, when insulating liquids are heated to elevated temperatures, some of their electrical and thermal characteristics (tanδ, σ, quality factor, QF, thermal conductivity, among others) undergo a change with time, and the change, even though of the minutest nature, may be reflected in the measurements. It is therefore desirable that the elapsed time necessary for the test specimen to attain temperature equilibrium with the TDC be kept to a minimum.

This section is organized as follows. First, the TDC and the electrical connections made with the measurement equipment are described; secondly, the Schering bridge method is explained; and, finally, the high-voltage experiments conducted are described.

### 3.1. Electrical Connection Diagram

Figure 6a shows a photograph of the TDC used in all the experiments. The high voltage should be connected to the handle on the inner cylindrical plate. In the measurement circuit, the V-potential should be connected to the metallic ring (electric guard) on the inner electrode (see Figure 7a,b).

The external cylinder should be insulated from the ground and connected to the measuring bridge. For this purpose, special connection cables were used. A clearance of several centimeters should be maintained between the HV connection and the metallic ring, which is connected to the V-potential, so that flashover does not occur between the HV-potential electrode and the ring. 

The test voltage should be raised to under 10 kV. The radial electrode spacing of the TDC is 6.7 mm. The sample should not break down under this voltage. Before the sample is tested, its temperature should be measured.

The TDC volume is about 1050 cm^3^. It should be filled until there is 2 cm of liquid above the top of the cylinder inside the TDC.

### 3.2. Schering Bridge

One of the most widely used methods to measure tan δ and $C_{p}$, with high precision, is the high-voltage Schering bridge [35].

The basic circuit layout is shown in Figure 7a. The capacitance $C_{p}$ and tan δ of a capacitor, or any capacitance of a sample, is measured by the bridge, comparing it to a standard capacitor $C_{N}$. This capacitor has very low losses, almost negligible, over a wide frequency range. It can be used for test voltages up to megavolts.

The high-voltage Schering bridge (Figure 7a) relies upon the measurement of the current ${\bar{I}}_{N}$ through the known reference capacitor $C_{N}=100\;\mathrm{pF}$ (SF6 insulated standard capacitor) and measurement of the current ${\bar{I}}_{t}$ through the unknown test object.$\;{\bar{I}}_{t}$ contains the currents ${\bar{I}}_{Cp}$ and ${\bar{I}}_{Rp}$. Comparing both currents, ${\bar{I}}_{N}$ and ${\bar{I}}_{t}$, the Schering bridge obtains the desired measurements.

Both branches are energized by an external HV-AC power source ${\bar{U}}_{Test}$. Both currents are measured by adjustable highly accurate shunts $R_{X}$ and $R_{N}$, and then digitized. By using IEEE 1394 “fire wire” data bus technology, each digitized value is time-stamped. With this technology, not only the values, but also the time information (phase shift) between ${\bar{I}}_{N}$ and ${\bar{I}}_{t}$, can be measured very rapidly and with a high degree of accuracy (see Figure 7a). The connection scheme of the metering equipment is shown in Figure 7b.

Shown below is the calculation of the values of capacity $C_{p}$ and resistance $R_{p}$. The resitence $R_{p}$ represents the oil losses. Both values are provided by the measurement system.

Applying Kirchhoff’s second law to the loop formed by the high-voltage source (HV-AC), the reference capacitor $C_{N}=100\;\mathrm{pF}$, and resistance $R_{N}$ used to measure the voltage ${\bar{U}}_{N}$, the following equation is obtained:(13)$${\bar{U}}_{Test}=\left[\frac{-j}{C_{N}\omega }+R_{N}\right]{\bar{I}}_{N}$$

In this equation, ω=2πf, where *f* = 50 Hz.

The current ${\bar{I}}_{N}=I_{N}|\underline{\varphi_{1}}$ is defined by its modulus $I_{N}$ and angle $\varphi_{1}$ taken with respect to the reference ${\bar{U}}_{Test}$, which is chosen as the phase origin.

The current ${\bar{I}}_{t}=I_{t}|\underline{\varphi_{X}}$ is fully defined in its modulus $I_{t}$ and angle $\varphi_{X}$, using the measurements performed as follows:(14)$$I_{t}=\frac{U_{X}}{R_{X}}$$
(15)$$\varphi_{X}=\varphi_{1}+\varphi_{m}$$
where $\varphi_{m}$ is the phase shift obtained between the voltages ${\bar{U}}_{N}$ and ${\bar{U}}_{X}$, using the measured times $t_{1}$ and $t_{2\;}$, in two consecutive homologous instants and provided by the measurement system (see Figure 7a).

Therefore, applying Kirchhoff’s second law to the loop formed by the source ${\bar{U}}_{Test}$, the resistance $R_{X}$, the capacitor $C_{p}$, and, considering the oil loss resistance $R_{p}$, the following is obtained:(16)$${\bar{U}}_{Test}=\left[R_{X}+\frac{R_{p}\frac{-j}{C_{p}\omega }}{R_{p}+\frac{-j}{C_{p}\omega }}\right]{\bar{I}}_{t}.$$

From this complex equation, $R_{p}$ and $C_{p}$ in the lumped parameter circuit of the TDC are unequivocally obtained.

### 3.3. High-Voltage Tests

This section presents the high-voltage laboratory, the connection diagram of the electrical equipment used in the experimental tests, the measurement procedure, and, finally, the experimental results obtained.

#### 3.3.1. Laboratory Description

The laboratory had two clearly separated areas. The high-voltage test and control rooms were where the low-voltage control and measurement equipment was located.

The test equipment, located in the high-voltage room, included a reference capacitor (Figure 8), transformer (Figure 9), and power supply (Figure 10).

Located in the control room was the OT 248 system operating terminal control equipment (Figure 11), which controlled the power supply, and the Schering bridge (Figure 12).

#### 3.3.2. Diagram of the Connections for the Electrical Test Equipment

The equipment connections were performed as indicated in the ASTM D 924-15 standard [10,12].

The OT 248 unit regulates the input power voltage. The power supply unit supplies to the system the input power voltage, in a low voltage (see Figure 13). Then, the transformer raises the single-phase voltage to the desired high voltage, always less than 10 kV, which is the TDC usage limit. The measuring bridge is connected to the reference capacitor *C_N_*, transformer, and, in turn, to the TDC (Figure 13).

#### 3.3.3. Temperature Measurement and Control Procedure

For safety, the transformer output was initially grounded. At the beginning of the test, this ground connection was removed. Then, the power supply unit was switched on. The desired maximum voltage level of 10 kV was specified in the OT 248 unit. Then, for the data collection with the Schering bridge, the voltage was gradually increased until the desired voltage was attained. 

The studied case began with 3 kV, reaching 8 kV in steps of 1 kV for each temperature. In the oil, temperatures of 23, 30, 40, 50, and 60 °C were successively recorded.

In this section, the control system of the TDC oil temperature is studied. The control objective was to progressively heat the oil and, subsequently, during the time that the test lasted, to maintain it a certain constant temperature.

For the reliable performance of dielectric strength tests, the TDC, as a standard device, cannot be modified. Therefore, it is not possible to immerse a thermometer into the oil during the test. It was necessary to obtain an indirect measurement of the oil temperature through four external thermometers placed on the external perimeter of the TDC. For this, the outer wall of the TDC was covered with neoprene, which served as thermal insulation.

DS18B20 waterproof temperature sensors were used [36]. A diagram of the location of these thermal sensors is shown in Figure 14. Their unique one-wire interface facilitated the communication with the devices. The DS18B20 can convert temperature to a 12-bit digital word in 750 ms, maximum. In addition, the DS18B20 can measure temperatures from −55 °C to +125 °C. It does not require an external power supply unit, as it is powered from the data line. Its stainless-steel probe head makes it suitable for any wet or harsh environment. Other features include the ranges of power supply from 3.0 to 5.5 V, and its accuracy (±0.5 °C over the range of −10 °C to +85 °C). The four thermal sensors were powered by a 3.2 V DC power supply.

The regulation system at present is an on–off controller, which feeds the heating resistances. If the oil temperature is lower than a certain set value, the electrical circuit is closed, and the intensity circulates through the four resistances connected in parallel. These gradually heat the oil until the established set-point is attained, at which point the current stops circulating.

The four resistances are each 22 Ω rated and are connected in parallel. They are powered by 20 V DC from an external power supply.

By means of an ESP-32 microcontroller, the temperature of the oil is registered and controlled. This is possible using the intensity control system of the four resistances. In addition, through the ESP-32, ambient temperature, relative humidity, and atmospheric pressure in the high-voltage room are recorded.

The ESP-32 microcontroller, working as a transmitter, is located in the high-voltage room. It transmits the measured temperature data in real time to the other paired ESP-32 microcontroller, working as a receiver, located in the control room. The two ESP-32 devices use Wi-Fi technology.

The transmitter and receiver boards are type ESP-32, 38-pin, dual-core, ultra-low-power, and Wi-Fi and Bluetooth compatible (see Figure 15). The ESP-32s are powered by AC rechargeable 5 V DC batteries. 

The room temperature, atmospheric pressure, and relative humidity were continuously monitored through the BME280 (3.3 V) digital sensor (see Figure 15). The average room temperature recorded during the experimental tests was 23 °C, the atmospheric pressure 986 mb, and the relative humidity 55%.

Figure 15 shows the layout of the four TDC temperature sensors, the BME280 digital sensor, and the microcontroller located in the high-voltage area. In addition, Figure 15 shows the microcontroller, display, and thermal set point regulator in the low-voltage area.

It was observed that the increase in oil temperature was approximately linear with respect to time. Given that 60 °C was attained in 36 min, it can be observed that about 6 min were sufficient to obtain an increase of 10 °C.

The dielectric strength test was performed with temperature jumps of 10 °C between steps. In the first 6 min, the oil temperature increased by 10 °C. To ensure a homogenous oil temperature distribution, we decided to wait an additional 24 min before performing a new test. During this time, this constant temperature was maintained by the control system, guaranteeing the homogeneity of the temperature field in the TDC during the 10 min tests. Schematically, Figure 16 represents the temporary heating process, temperature homogenization, and tan δ test.

To verify that the temperature, indirectly measured by external thermometers, which matched the actual oil temperature, thermographs of the oil surface were taken with an FLIR E54 camera [37].

These thermographs were taken in tests prior to the high-voltage test. For this purpose, the TDC cover was removed, allowing the immersion of the DS18B20 digital thermometer in the oil to provide a direct measurement of the oil’s temperature.

The FLIR E54 thermal camera offers the resolution and sensitivity needed for the basic monitoring and inspection of electrical/mechanical devices. Its thermal detector accurately measures temperatures up to 650 °C. The FLIR E54 also shows the maximum/minimum temperatures in an area of its screen.

In infrared mode, its resolution is 320 × 240 pixels. The thermal sensitivity or noise equivalent temperature difference (NETD) is less than 40 mK at 30 °C. The NETD is equivalent to the smallest difference in temperature that the camera can measure without being attributed to its own noise. Thermal sensitivity is usually described in °C or mK (thousandths of a Kelvin). High-end thermal-imaging cameras offer sensitivities around 30 mK [38].

The FLIR E54 thermal camera is accurate to ±2 °C or ±2% of the reading. It is a 5 MP digital camera with an integrated 4-inch photo/video LED illumination, 640 × 480 pixels, and LCD touch screen with auto-rotate and removable SD card as storage media.

Figure 17a shows the thermograph of the oil-filled TDC at 62 °C. It can be observed that the horizontal surface of the oil has an approximately uniform temperature, except in the vicinity of the electrical resistance that has a slightly higher temperature.

#### 3.3.4. Test Results

Throughout the experimentation, the ASTM D 924-15 standard [12] was followed. The parameters obtained in the oil tests were the following (see Figure 6b and Figure 7a):

*R_p_* = Parallel resistance of the equivalent circuit of lumped parameters.*C_p_* = Parallel capacity of the equivalent circuit of lumped parameters.$\bar{I}{\;}_{N}$ = Electric current passing through the reference capacitor *C_N_* of 100 pF.${\bar{I}}_{Cp}$ = Electric current passing through the capacitor *C_p_*.*tan δ* = Ratio between the electric current ${\bar{I}}_{Rp}$ that passes through the resistance *R_p_* and the current ${\bar{I}}_{Cp}$, as defined in Equation (9).QF = Quality factor. Ratio between the energy stored in the electric field of the real capacitor *C_p_* divided by the energy dissipated by the resistance *R_p_* in a period of time at the operating frequency.

The tan δ and QF values are parameters of great importance as they characterize the dielectric behavior of the oil. Both allow for the evaluation of the dielectric losses in the insulation and thus detect, at an early stage, signs of aging in the insulation or accessories, such as bushings.

$R_{p}$ and $C_{p}$, the resistance and equivalent capacity of the TDC and ${\bar{I}}_{N}$ and ${\bar{I}}_{t}$, the Schering measurement bridge currents, allow us to obtain numerically ε and σ. This was conducted in Section 4.

In the present work, the values of these parameters were determined for voltages between 3 and 8 kV, applied in successive steps of 1 kV for different temperatures. Temperatures of 30, 40, 50, and 60 °C were recorded in the oil, in addition to room temperature.

Given the large amount of data recorded, the results are graphically presented in this paper (Figure 18, Figure 19, Figure 20, Figure 21, Figure 22 and Figure 23) as opposed to the use of dense numerical tables.

Figure 18 plots tan δ against test voltage ${\bar{U}}_{Test}$ for the five different oil temperatures *T*. It can be observed that tan δ increases with ${\bar{U}}_{Test}$ and *T*. This was expected as electrical losses increase with applied voltage.

Figure 19 plots QF against ${\bar{U}}_{Test}$ for the five different *T*. It can be observed that QF decreases as ${\bar{U}}_{Test}$ and *T* increase. Therefore, the two plots are consistent, because an increase in tan δ is detrimental, as is a decrease in QF.

Figure 20 plots ${\bar{I}}_{N}$ against ${\bar{U}}_{Test}$ for the five different *T*. It can be observed that ${\bar{I}}_{N}$ is independent of *T*, since the five curves coincide, and that it varies linearly with ${\bar{U}}_{Test}$.

Figure 21 plots ${\bar{I}}_{t}$ against ${\bar{U}}_{Test}$ for the five different *T*. It can be observed that ${\bar{I}}_{t}$ is independent of *T*, since the five curves coincide, and that it varies linearly with ${\bar{U}}_{Test}$. The behavior is similar to that shown by ${\bar{I}}_{N}$.

Figure 22 plots $R_{p}$ against ${\bar{U}}_{Test}$ for the five different *T*. It can be observed that $R_{p}$ decreases as ${\bar{U}}_{Test}$ and *T* increase. This is consistent with what was observed for tan δ, since $R_{p}$ decreases, increasing the electrical losses.

The well-known Arrhenius’s equation relates σ to *T* through the following analytic equation [5]:(17)$$\sigma \left(T\right)=\left[A\cdot exp\left(-\frac{E_{ac}}{K\cdot T}\right)\right],$$
where *A* is a constant related to ion mobility, $E_{ac}$ is the activation energy, and *K* is the Boltzmann constant. Since the electrical resistivity *ρ* is the inverse of *σ*, this equation shows that $R_{p}$ decreases with increasing temperature, as seen above.

Figure 23 plots $C_{p}$ against ${\bar{U}}_{Test}$ and *T*. It can be observed that $C_{p}$ is independent of ${\bar{U}}_{Test}$ and that it decreases with *T*.

## 4. Fit of the Effective Parameters Using the CM and NSGA-II

In this section, a numerical method is proposed to accurately determine σ and ε in an insulating mineral oil from the data obtained in the previous section. The CM and NSGA-II methods were used.

In Section 4.1, a new numerical tool is provided to accurately obtain the effective electrical parameters of transformer insulating oils, *σ* and *ε*. Its accuracy is improved with respect to existing analytical equations as a distributed parameter model is used together with GA-based optimization.

In Section 4.2, the Results and Discussion are presented.

### 4.1. Methodology

The procedure followed by the NSGA-II algorithm, represented in Figure 24, is summarized below [39]:

Step 1. An initial random population of size N is generated, respecting the ranges of the variables. In this particular problem, the physical properties to be determined ($\varepsilon_{r}-\sigma$) are equivalent to a chromosome. In addition, the values of the objective functions *g_1_* and *g_2_* are evaluated, which are the errors with respect to a reference of the experimental values *C_p_* and *R_p_* for that combination of variables.Step 2. The initial population is classified based on the values of the objective functions to generate several non-dominated fronts. Each member of each front is assigned a fitness value, called a rank.Step 3. The crowding distance of each member in each front is calculated. The fronts are ordered according to their objective values.Step 4. According to the range and crowding distance, the main population is ordered by selecting the nest individuals through a binary tournament competition.Step 5. From those selected in the previous step, the original population is crossed and mutated to generate an offspring population.Step 6. The initial population of parents and offspring obtained from Step 5 is combined to generate a population size of *2N*. From this last selection, elitism is performed to select the best population of size N, according to the value of the objective functions.Step 7. The stopping criterion of the iteration in progress is checked, which consists of verifying if the maximum number of generations has been attained. If this condition is not reached, steps 2 to 7 are repeated.

The objective functions used in the NSGA-II model define the error that is produced when we are searching for the final solutions of the iterative problem, σ and $\varepsilon_{r}$. In this way, we fitted the experimental data obtained in Section 3 to the numerical solutions obtained through CM.

The paper followed the ASTM D 924-15 standard where effective electrical parameters, including the tanδ of the oil, were obtained. However, note that tanδ is not used directly in the numerical model developed in this section to obtain the effective parameters, σ and ε. The parameters that are directly involved in the model are $R_{p}$, the parallel resistance of the equivalent circuit of lumped parameters, and $C_{p}$, the parallel capacity of the equivalent circuit of lumped parameters.

Figure 24 shows a flow diagram of the iterative global process until the effective parameters of the dielectric oil, $\varepsilon_{r}$ and σ, are finally obtained. The flow diagram begins with the external conditions *T* and ${\bar{U}}_{Test}\;$applied to the dielectric. Once the measurements have been accurately made with the Schering bridge (see Section 3.2), the experimental parameters $C_{p}\left(T,{\bar{U}}_{Test}\right)$ and $R_{p}\left(T,{\bar{U}}_{Test}\right)$ are obtained. These two values are used as references in the objective functions *g_1_* and *g_2_*:(18)$$g_{1}=Error_{C}\left(\%\right)=\frac{C_{p}\left(T,{\bar{U}}_{Test}\right)-C_{P}^{i}}{C_{p}\left(T,{\bar{U}}_{Test}\right)}*100,$$
(19)$$\;g_{2}=Error_{R}\left(\%\right)=\frac{R_{p}\left(T,{\bar{U}}_{Test}\right)-R_{P}^{i}}{R_{p}\left(T,{\bar{U}}_{Test}\right)}*100,$$
where $C_{P}^{i}$ y $R_{P}^{i}$ are the values obtained in the post-processing stage when Equation (2) is solved, and which are calculated through Equations (4), (5), (11), and (12).

Note that the objective function $g_{2}$ is directly linked to σ. This is because σ is the inverse of ρ, and $R_{p}$ is directly proportional to ρ.

### 4.2. Results and Discussion

The experimental input data in the flowchart of Figure 24 were obtained, as described in Section 3.3, from a total of 30 experiments with five temperatures and six different voltages (23–60 °C and 3–8 kV).

Table 2 summarizes twelve of these numerical optimization experiments, allowing the visual verification of the proposed method and the results obtained for the effective parameters σ and $\varepsilon_{r}$. The parameters of the GA common to these calculations are shown in Table 3.

C and C++ were used as the programming language. The source code of each program for the MOGA was obtained from the Kanpur Genetic Algorithms Laboratory [40]. The authors of the present paper developed the program for the solution of Equation (2) in C++.

A real codification was chosen for each chromosome $\left(\varepsilon_{r}-\sigma \right)$. In general, the precision is better than binary codification. It can be improved by adding more bits, but this increases the simulation time [19,21]. As the variable of the parameter space of an optimization problem is continuous, a real coded GA is possibly preferred [41].

As previously mentioned, the results of 12 of the 30 experiments are presented, named GA1 to GA12. They have in common the parameters in Table 3 and differ, as indicated in Table 2, in terms of the number of objective functions used: either one function using the weighted sum (WS) method, or two functions using the method based on the concept of the Pareto front (PF).

The PF method is based on minimizing two error functions, *g_1_* and *g_2_*. The WS method minimizes a single average function:(20)$$g_{SP}=\frac{1}{2}\left(Error_{C}\left(\%\right)+Error_{R}\left(\%\right)\right).$$

The number of generations performed in the majority of the calculations was 20. However, in the GA11 and GA12 calculations, the number of generations was increased to 100 in order to verify the convergence of the results.

In step 1, an initial random population of size N equal to 40 individuals was generated. The execution time was approximately 56 min for 100 generations. In this problem, each individual of the population was equivalent to a chromosome, which was constituted by the union of the physical properties to be determined, $\varepsilon_{r}$ and σ [21].

For the codification of the chromosome, the algorithm needs to know the minimum and maximum intervals of these variables. For σ, which is part of the chromosome, the values considered are a lower limit of 0.15 pS/m, which is a very low value, and an upper limit of 80 pS/m, which is a very high value. All the experiments conducted were between those two limits. In the same way, the lower and upper limits of $\varepsilon_{r}$ were 1 and 4, respectively. One corresponds to the air and four is the maximum value observed in the literature.

Figure 25 shows the distribution of results for the 20 generations and a population of 40 individuals corresponding to the GAI experiment (Table 2). In Figure 25a, $\varepsilon_{r}$ is plotted against the error *Error_C_(%)* for all generations. Likewise, the optimal value is represented, which is 2.1, for $\varepsilon_{r}$, with an error close to 0.

It can be observed how the distribution of the results has the shape of an arrow that points to and converges with the optimal value. In Figure 25b, the distribution of σ is plotted against the error *ErrorR(%)*. This distribution, in the form of an arrow, similarly points to the optimal value (42.3 pS/m), which coincides with the value obtained by an exhaustive search method. The values, represented in green in both figures, correspond to the best generation.

In order to check the convergence of the results of the GA1 experiment, based on the Pareto optimum, the number of generations was increased from 20 to 100, while the other GA parameters (Table 3) were unchanged. The results are shown in Figure 26. The two curves represented in Figure 26a show the best result obtained in the search for any generation and the average of the results of the entire population in each generation. The two curves are practically coincident from generation 20, indicating that, with only 20 generations, an optimal value of $\varepsilon_{r}$ is obtained.

However, in Figure 26b, it is observed that, even though the corresponding values of σ converge to a minimum error for the best values, their means do not converge. This is because the Pareto front coincides with the x–y axes. Therefore, a distribution on this front occurs. This corresponds to the ordinate axis, and, for the objective function *Error_R_(%)* also. These distributions are uniform.

A new numerical experiment was conducted based on the WS method (GA12 in Table 2) with the aim of improving the convergence of *σ* of the population mean. In Figure 27a, the convergence of $\varepsilon_{r}$ is represented. Starting from generation 20, it can be observed how the best and mean values practically coincide. In Figure 27b, using WS as an objective function, a convergence of σ occurs between the best and mean values from generation 20. The latter is explained in Figure 28b, where it is observed that the weighted average error, from generation 20, is very small.

Figure 29 plots $\varepsilon_{r}$ against ${\bar{U}}_{Test}$ for the five different oil temperatures *T*. It can be observed that $\varepsilon_{r}$ is independent of ${\bar{U}}_{Test}$ and decreases as *T* rises.

Figure 30 plots σ against ${\bar{U}}_{Test}$ for the five different oil temperatures *T*. It can be observed that σ increases with ${\bar{U}}_{Test}$ and *T*.

Then, we presented a comparison of the results obtained with the approximate analytical method presented in [5]. Figure 31 and Figure 32 show the average errors of ε and σ that are committed to this approximate method, which does not take into account the geometry of the TDC.

After obtaining the physical parameters σ and $\varepsilon_{r}$, an evaluation of σ vs. *T* was performed using the Arrhenius equation.

As explained in Section 3.3.4, the data obtained from σ depend on the temperature. The data were fitted to the Arrhenius exponential equation. Following the indications of [8], the adjusted coefficients of $E_{ac}$ and A were obtained on the basis of this equation; the data of σ were obtained in the optimization process and *T*. These coefficients depend on the applied voltage, since σ also depends on it, as can be observed in Figure 30. The values obtained for $E_{ac}$ and A are summarized in Table 4. It can be observed how both coefficients decrease with ${\bar{U}}_{Test}$.

Figure 33 shows the experimental data obtained for electrical conductivity as a function of temperature. These are represented by black dots.

In addition, based on the coefficients $E_{ac}$ and A, the functions obtained from the Arrhenius equation are represented for each applied ${\bar{U}}_{Test}$, from 3 to 8 kV. These curves are represented as solid lines.

The knowledge of these functions is important, since it allows us to obtain the values of σ at different temperatures and voltages from those measured experimentally. It can be observed that the experimental data fit the theoretical Arrhenius curves with high accuracy.

## 5. Conclusions

In this paper, an experimental NDT analysis of the quality of electrical insulating oils was conducted using a combination of dielectric loss and capacitance measurement tests. The transformer oil corresponded to a fresh oil sample. Based on the standard ASTM D 924-15 (standard test method for dissipation factor and relative permittivity of electrical insulating liquids), effective electrical parameters, such as tan δ of the oil, were obtained.

In addition, a numerical method was proposed to accurately determine, on the basis of the data obtained in the experimental analysis, the effective electrical resistivity, σ, and effective electrical permittivity, ε, of an insulating mineral oil for the purpose of fault detection and diagnosis. The cell method and non-dominated sorting in genetic algorithm-II (NSGA-II) for multi-objective optimization were used, showing improved accuracy compared to the existing analytical equations.

The results of the calculations and optimizations conducted show that the best method for adjusting the properties corresponds to a WS objective function, which is the weighted average, at 50% between the functions *Error_C_(%)* and *Error_R_(%)*. The convergence between the best values of the population and the mean of the population occurs from generation 20.

As the experimental data were collected in a high-voltage domain, wireless sensors were used to measure, transmit, and monitor the electrical and thermal magnitudes.

## Figures and Tables

**Figure 1 sensors-23-01685-f001:**
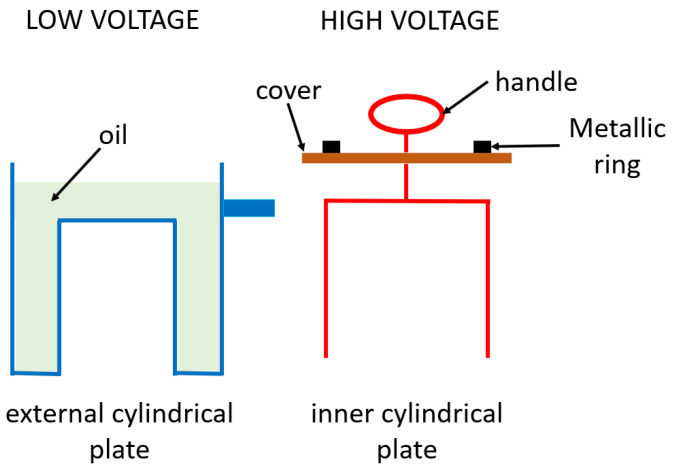
Schematic diagram of the TDC understood as a cylindrical capacitor.

**Figure 2 sensors-23-01685-f002:**
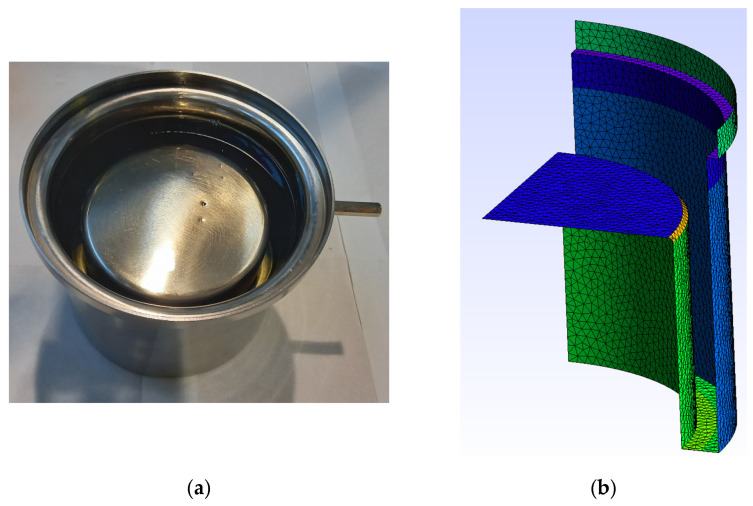
(**a**) Low-voltage external cylindrical plate. (**b**) Two-dimensional mesh generated with Gmsh.

**Figure 3 sensors-23-01685-f003:**
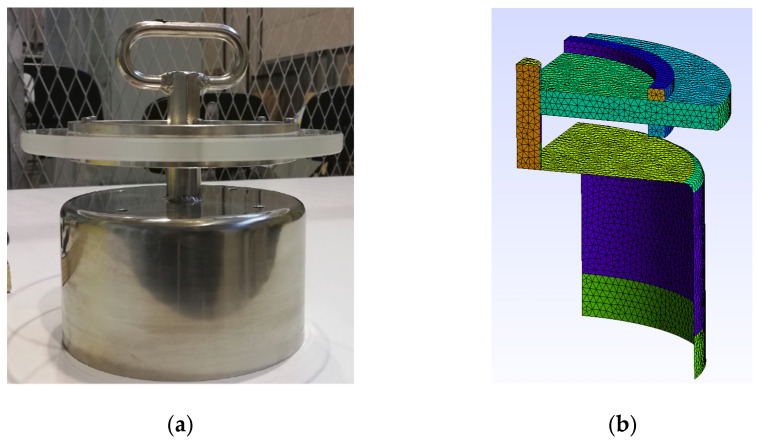
(**a**) High-voltage inner cylindrical plate. (**b**) Mesh generated with Gmsh.

**Figure 4 sensors-23-01685-f004:**
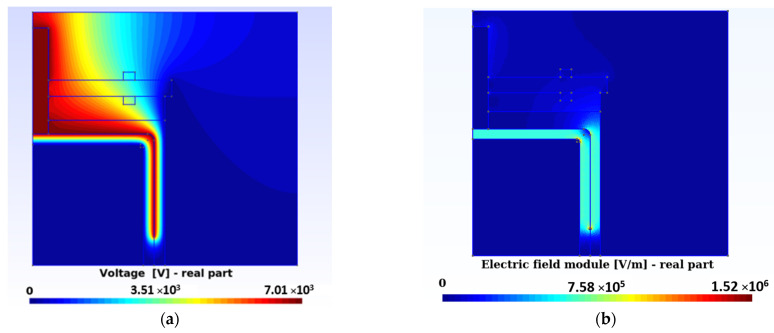
(**a**) Voltage distribution at TDC without oil, with 14,165 nodes and solved with Equation (2) using 2D-CM. (**b**) Distribution of the electric field strength modulus corresponding to the numerical experiment AG3 in Section 4.1.

**Figure 5 sensors-23-01685-f005:**
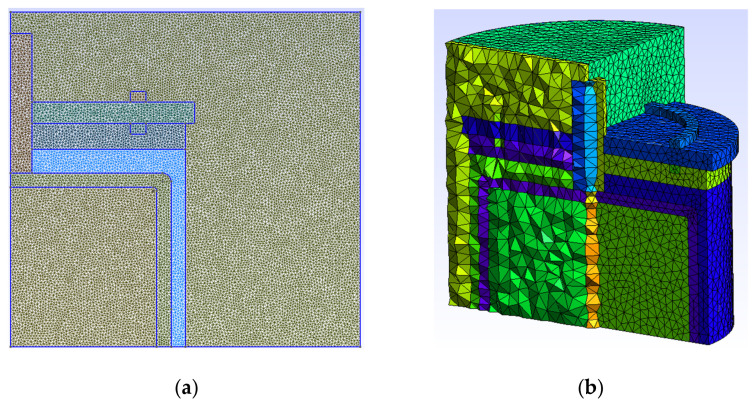
Comparison of 2D versus 3D meshing. (**a**) Two-dimensional mesh. Number of nodes: 14,165. Number of triangles: 27,842. (**b**) Three-dimensional mesh. Number of nodes: 20,269. Number of tetrahedra: 108,154.

**Figure 6 sensors-23-01685-f006:**
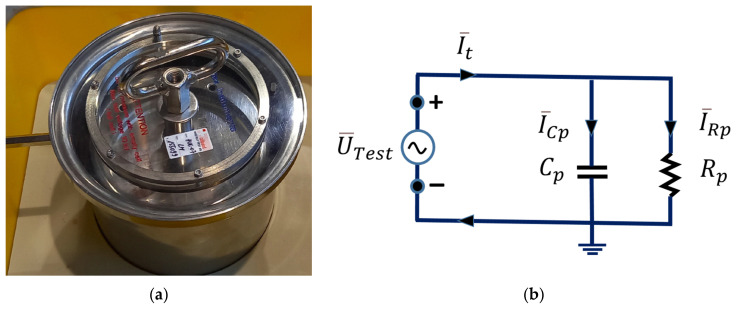
TDC lumped parameter model. (**a**) TDC with reference capacitance of 109 pF. (**b**) Equivalent circuit of lumped parameters of the TDC.

**Figure 7 sensors-23-01685-f007:**
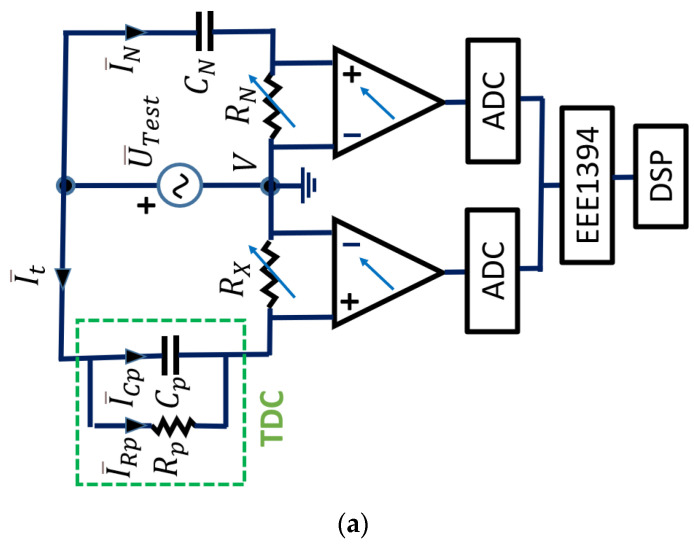
(**a**) Detailed electrical circuit of the Schering bridge, including the TDC. (**b**) Connection diagram of the TDC and Schering bridge.

**Figure 8 sensors-23-01685-f008:**
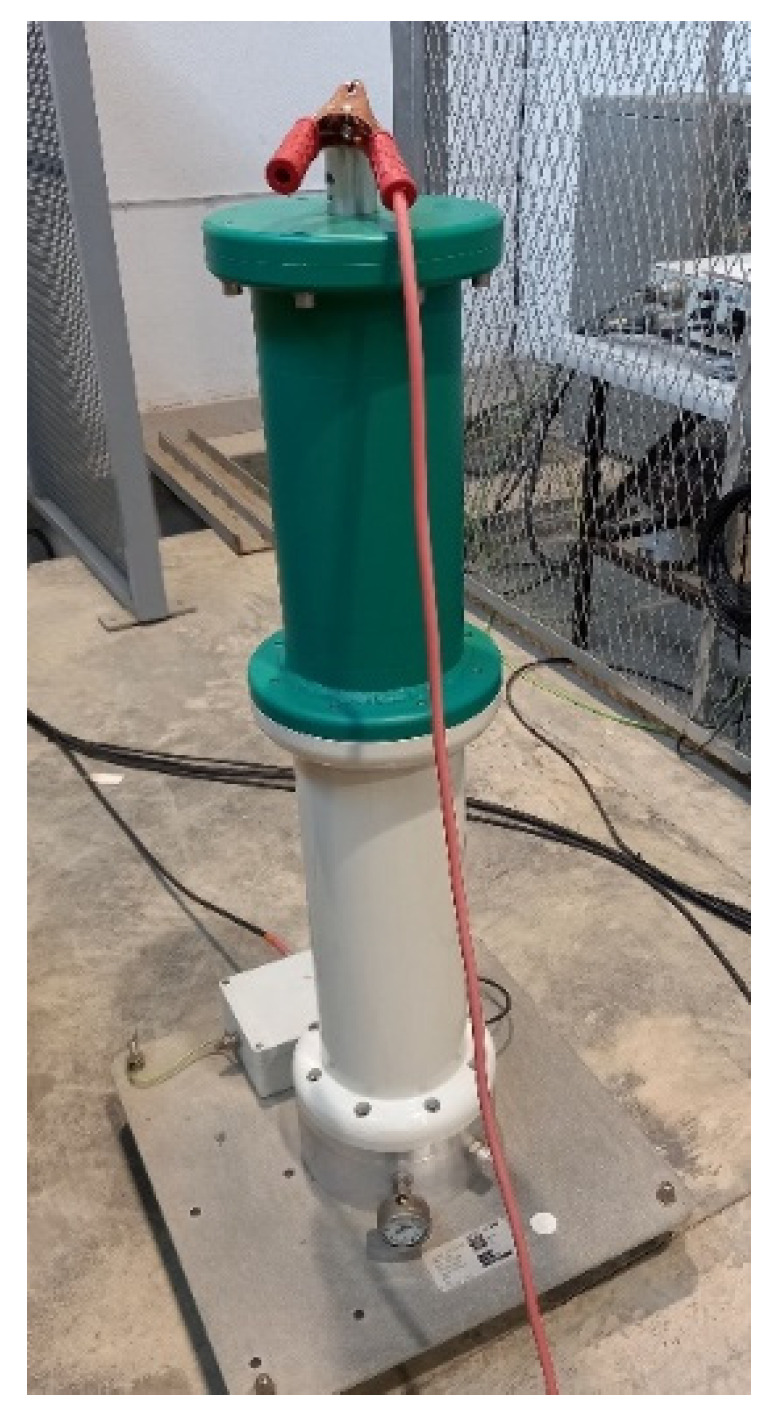
Reference capacitor, Haefely Test AG, rated capacitance 100 pF, rated voltage 100 kV, dissipation factor 1 × 10^−5^.

**Figure 9 sensors-23-01685-f009:**
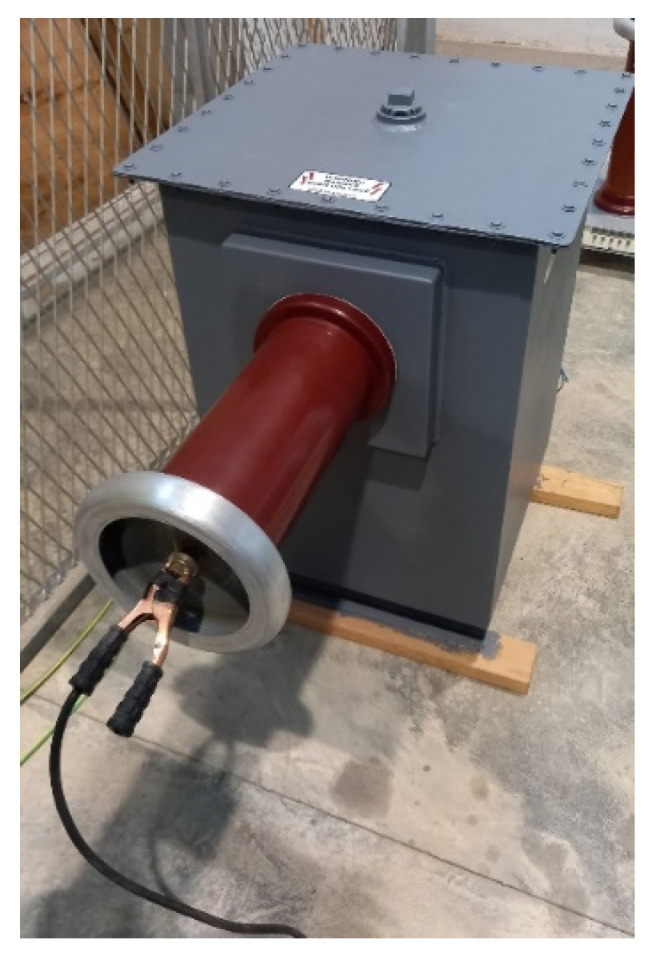
HV test transformer model HHT7-10T-220, Hipotronics-Hubbell, rated input 230 V AC, 10 KVA, rated output 100 KVA, 100 mA, freq. 50 Hz.

**Figure 10 sensors-23-01685-f010:**
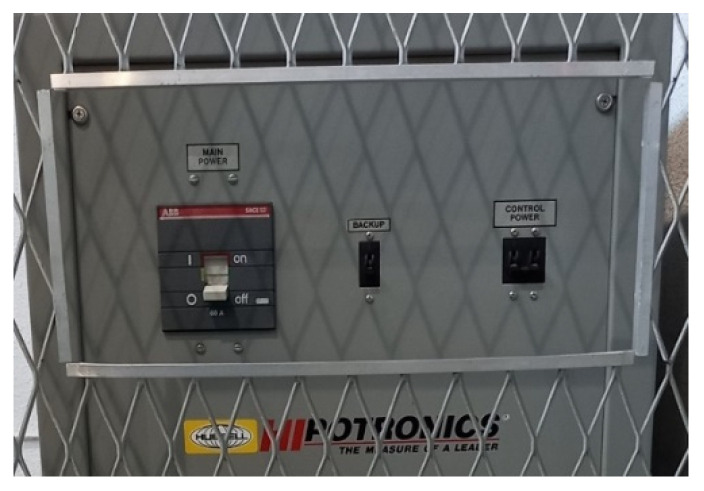
Power supply for the regulation of the output voltage, from 0 to 230 V.

**Figure 11 sensors-23-01685-f011:**
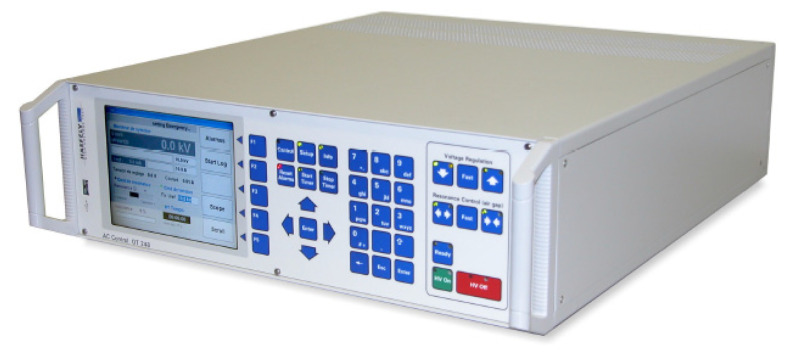
Operating terminal OT 248 system, Tettex-Haefely test AG.

**Figure 12 sensors-23-01685-f012:**
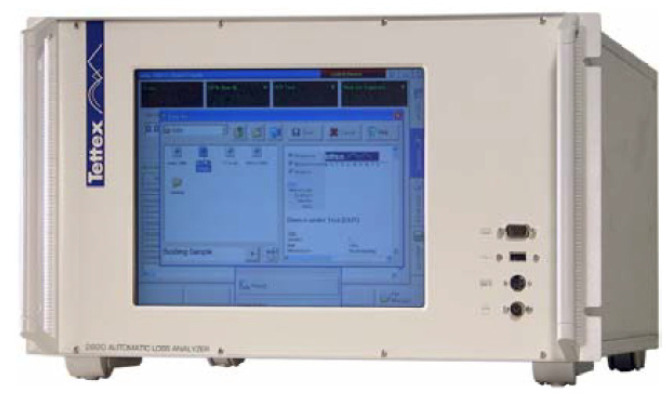
2820 Automatic capacitance, inductance, and tan δ measurement bridge, 0 to 1000 kV, Tettex-Haefely test AG. Schering bridge.

**Figure 13 sensors-23-01685-f013:**
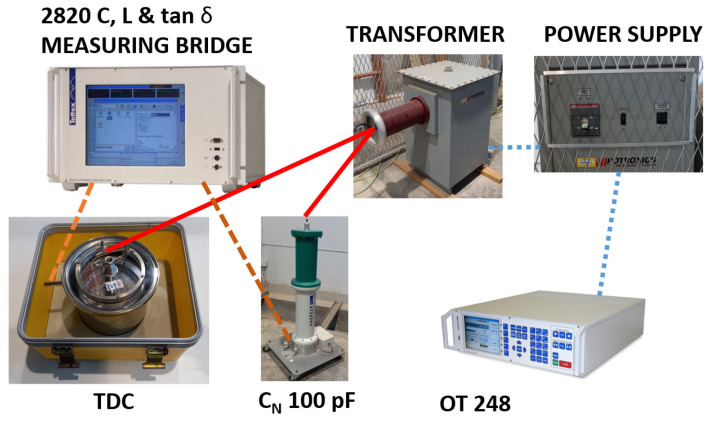
Electrical equipment connection diagram.

**Figure 14 sensors-23-01685-f014:**
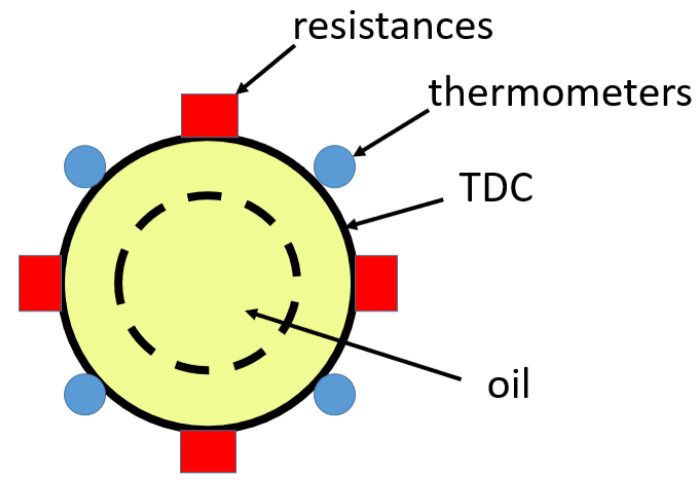
Diagram of the arrangement of resistors and thermometers in the TDC.

**Figure 15 sensors-23-01685-f015:**
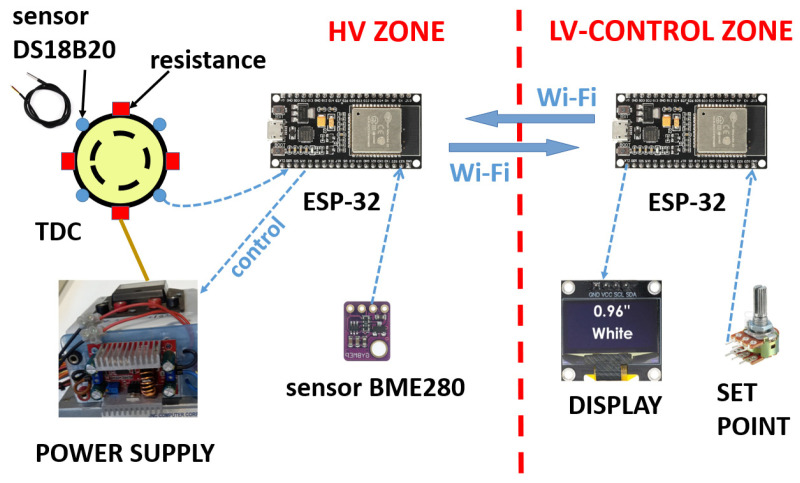
Connection diagram of the TDC measurement and control system.

**Figure 16 sensors-23-01685-f016:**
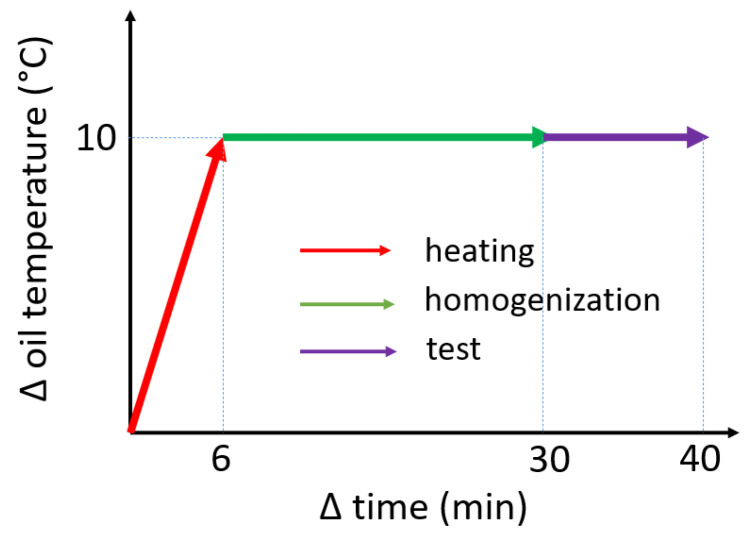
Heating, temperature homogenization, and test times for the global test.

**Figure 17 sensors-23-01685-f017:**
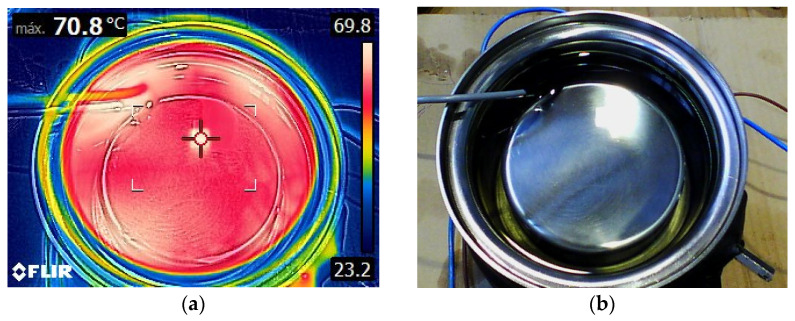
(**a**) Thermograph of the deposit of the TDC. The oil has a uniform temperature when the thermal set point is reached (62 °C in this case). (**b**) Photograph of the TDC container without its lid. Above left: the DS18B20 waterproof temperature sensor.

**Figure 18 sensors-23-01685-f018:**
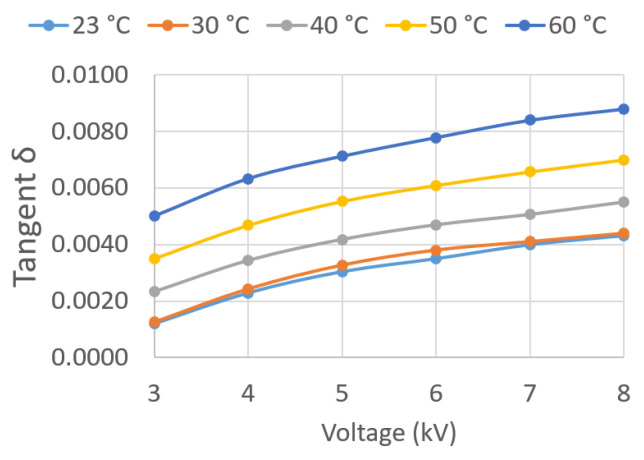
tan δ vs. applied voltage ${\bar{U}}_{Test}$.

**Figure 19 sensors-23-01685-f019:**
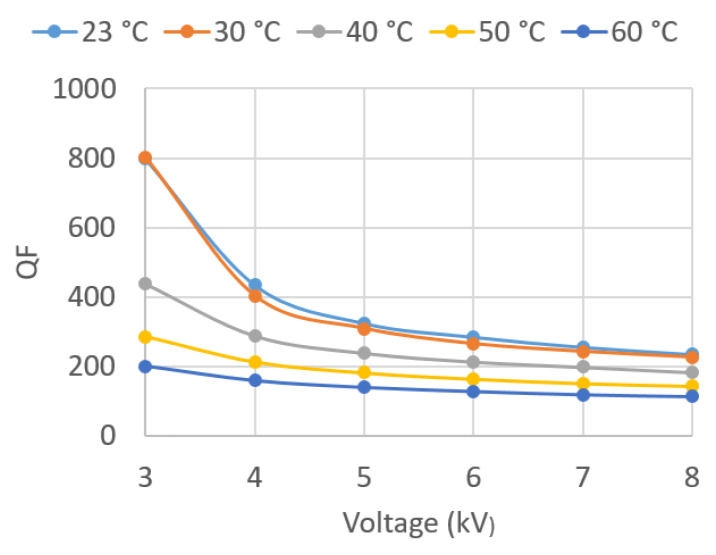
QF vs. applied voltage ${\bar{U}}_{Test}$.

**Figure 20 sensors-23-01685-f020:**
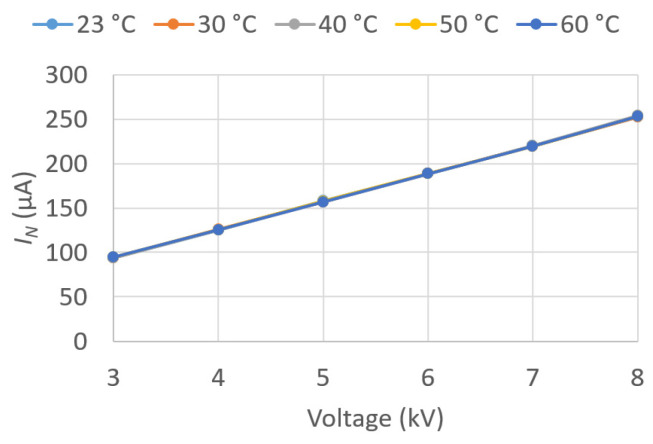
$I_{N}$ vs. applied voltage ${\bar{U}}_{Test}$.

**Figure 21 sensors-23-01685-f021:**
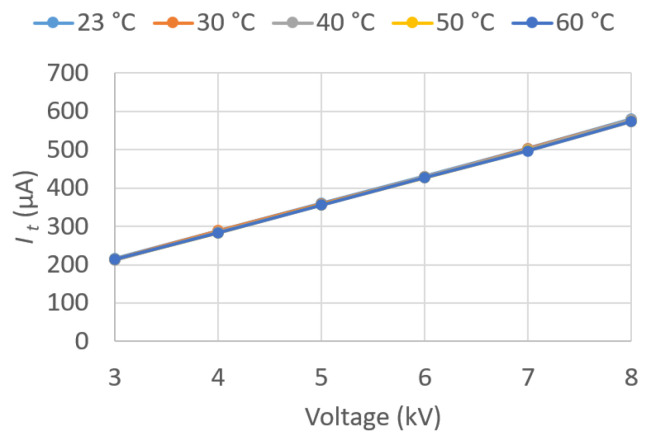
$I_{t}$ vs. applied voltage ${\bar{U}}_{Test}$.

**Figure 22 sensors-23-01685-f022:**
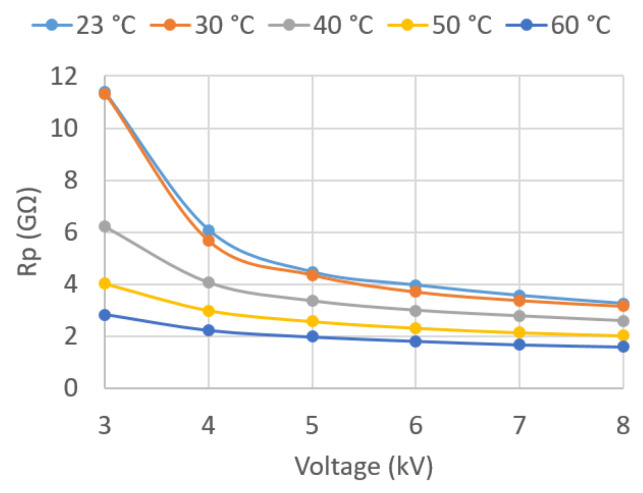
$R_{p}$ vs. applied voltage ${\bar{U}}_{Test}$.

**Figure 23 sensors-23-01685-f023:**
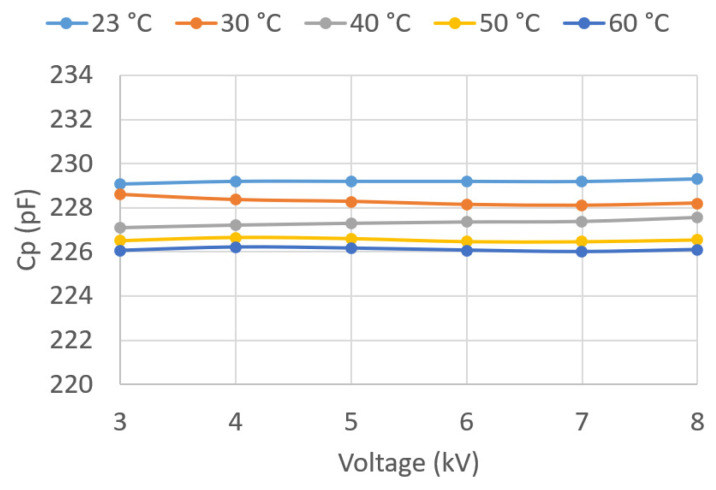
$C_{p}$ vs. applied voltage ${\bar{U}}_{Test}$.

**Figure 24 sensors-23-01685-f024:**
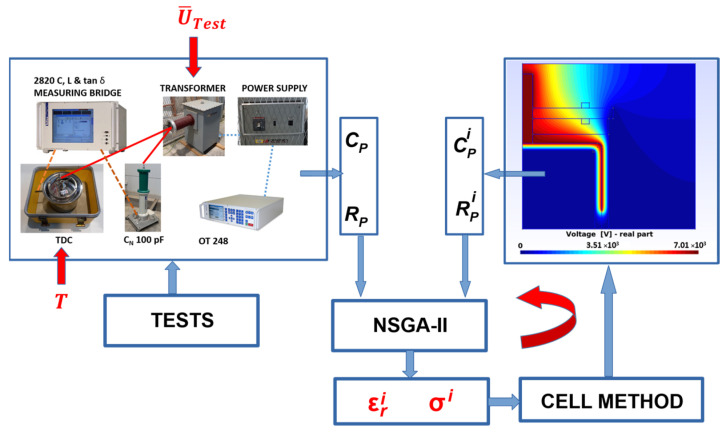
Flow diagram of the determination of the effective electrical parameters σ and $\varepsilon_{r}$ using NSGA-II and CM on the basis of the experimental data.

**Figure 25 sensors-23-01685-f025:**
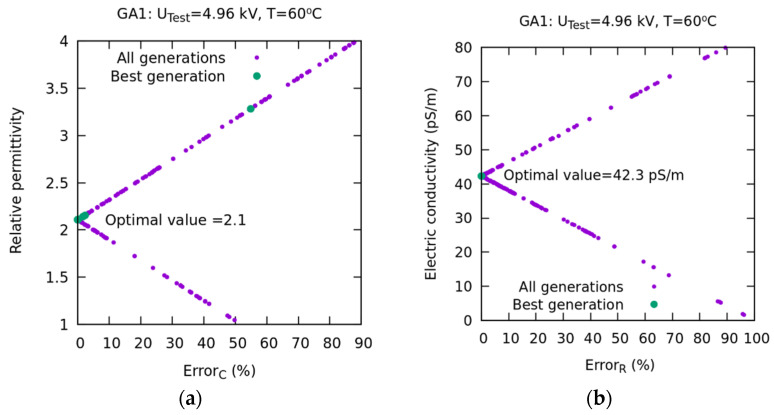
PF-based method. Results of the GA1 numerical experiment. (**a**) $\varepsilon_{r}$ vs. *Error_C_(%)* for all generations. The optimum value is 2.1. (**b**) σ vs. *Error_R_(%)* for all generations. The optimum value is 42.3 pS/m.

**Figure 26 sensors-23-01685-f026:**
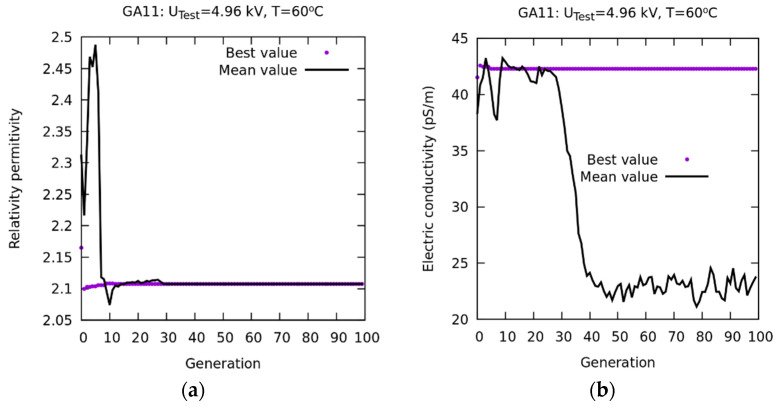
PF-based method. Results of the GA11 numerical experiment. (**a**) Convergence of $\varepsilon_{r}$. (**b**) Convergence of σ.

**Figure 27 sensors-23-01685-f027:**
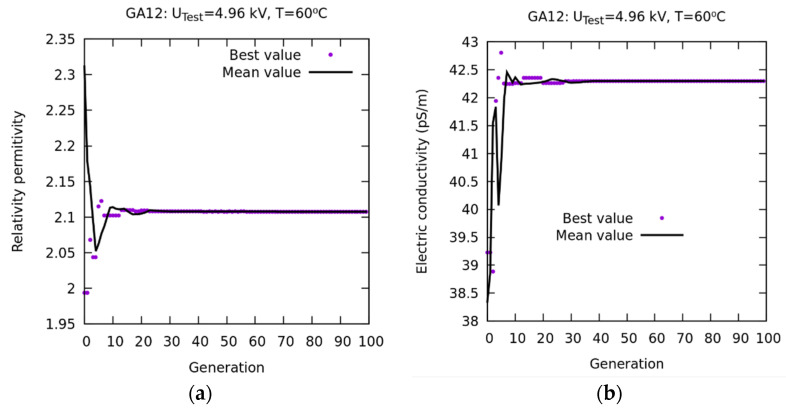
WS-based method. Results of the GA12 numerical experiment. Convergence for a single objective function. (**a**) Convergence of $\varepsilon_{r}$. (**b**) Convergence of σ.

**Figure 28 sensors-23-01685-f028:**
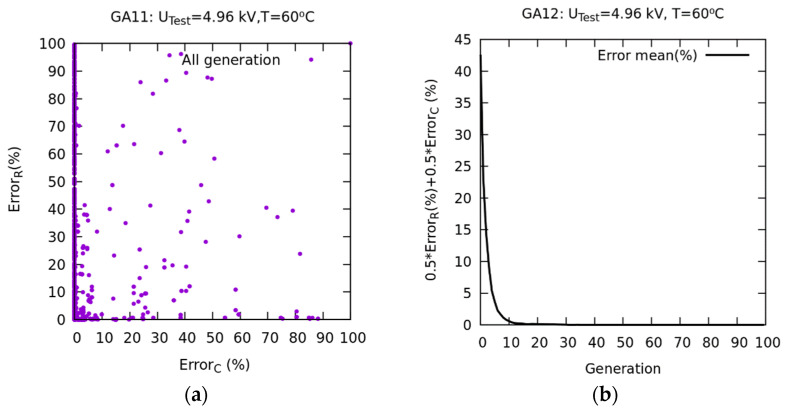
(**a**) Representation of the PF for two objectives. (**b**) Convergence of the mean error for one objective.

**Figure 29 sensors-23-01685-f029:**
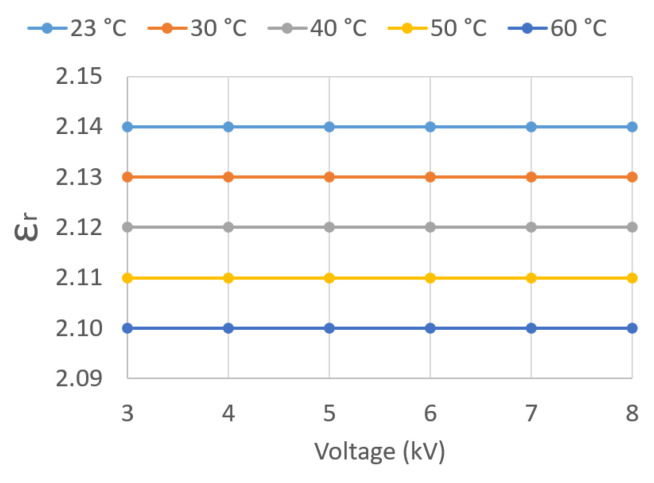
$\varepsilon_{r}$ vs. test applied voltage ${\bar{U}}_{Test}$.

**Figure 30 sensors-23-01685-f030:**
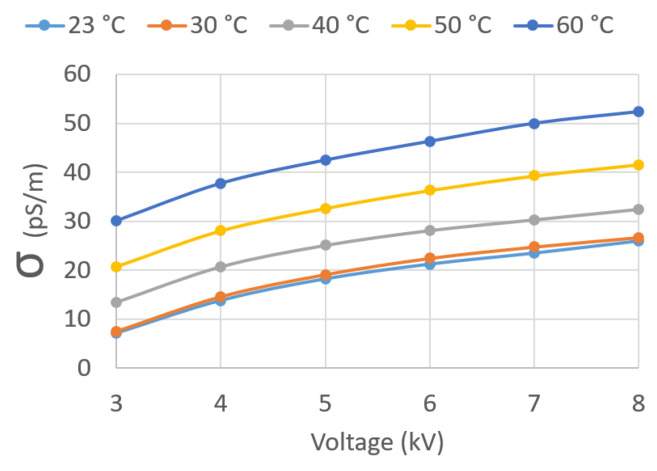
σ vs. test applied voltage ${\bar{U}}_{Test}$.

**Figure 31 sensors-23-01685-f031:**
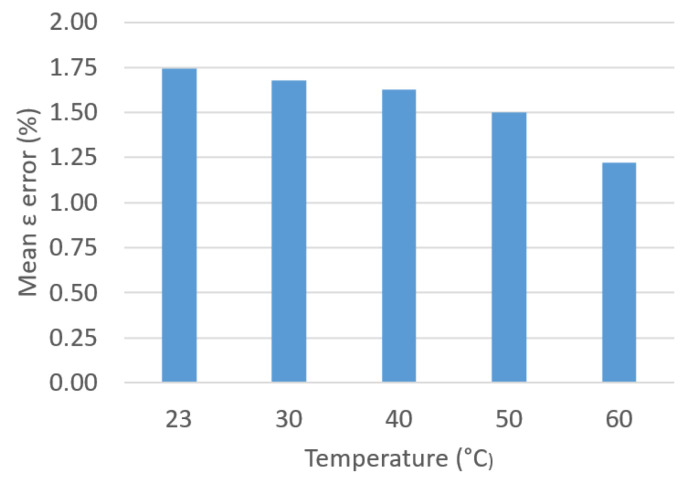
Mean ε error (%) vs. *T*.

**Figure 32 sensors-23-01685-f032:**
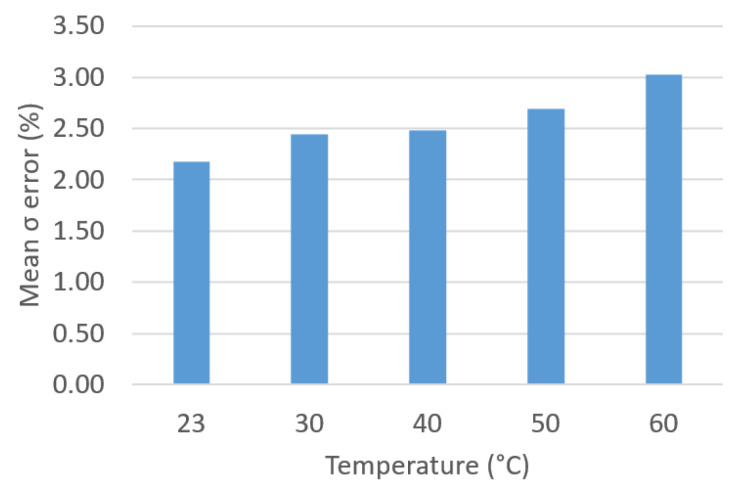
Mean σ error (%) vs. *T*.

**Figure 33 sensors-23-01685-f033:**
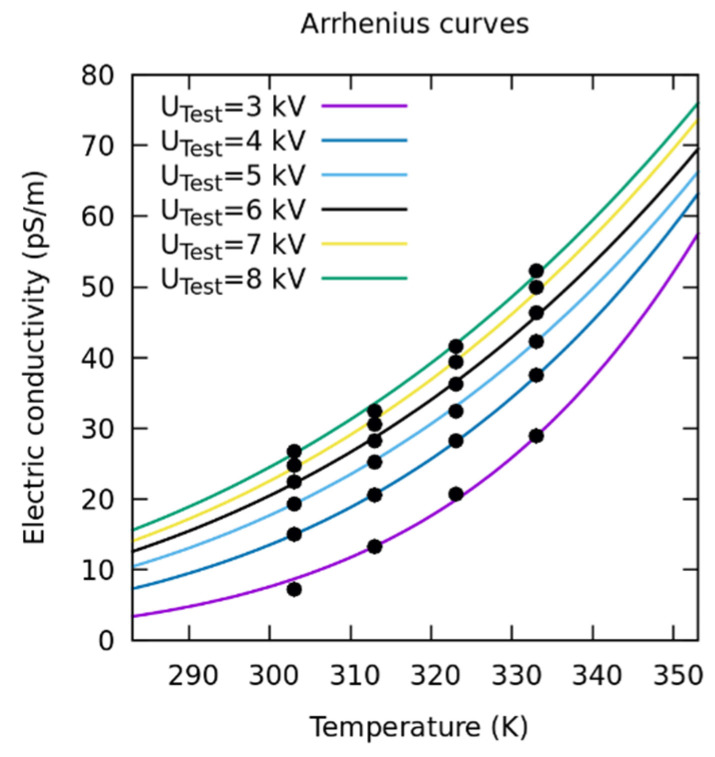
σ vs. *T*. The GA-obtained values, represented by dots, conform to the theoretical Arrhenius curves represented by solid lines.

**Table 1 sensors-23-01685-t001:** Properties of the Haefely Hipotronics 6835 TDC.

Property	Value
Empty capacitance	About 109 pF
Radial electrode spacing	6.7 mm
Quantity of liquid	About 1050 cm^3^
Max. test voltage	10 kV, 50/60 Hz
Dimensions	Ø 178 × H 190 mm

**Table 2 sensors-23-01685-t002:** Summary of calculations obtained by NSGA-II.

Experiment Name	Number of Objective Functions and Method	Number of Generations	*U_Test_* (kV)	*T* (°C)	$\varepsilon_{r}$	σ (pS/m)
GA1	2-PF	20	4.96	60	2.10	42.3
GA2	2-PF	20	8.05	60	2.10	52.3
GA3	2-PF	20	3.00	50	2.11	20.7
GA4	2-PF	20	6.01	50	2.11	36.3
GA5	2-PF	20	3.96	40	2.12	20.6
GA6	2-PF	20	8.11	40	2.12	32.4
GA7	2-PF	20	2.96	30	2.13	7.2
GA8	2-PF	20	8.04	30	2.13	26.7
GA9	2-PF	20	5.17	23	2.14	18.6
GA10	2-PF	20	6.02	23	2.14	21.2
GA11	2-PF	100	4.96	60	2.10	42.3
GA12	1-WS	100	4.96	60	2.10	42.3

**Table 3 sensors-23-01685-t003:** Execution parameters of genetic algorithms.

Parameter	Value
Population size	40
Number of real variables	2
Lower limit of real variable 1 [*σ*]	0.152 pS/m
Upper limit of real variable 1 [*σ*]	80.0 pS/m
Lower limit of real variable 2 [*ε_r_*]	1.0
Upper limit of real variable 2 [*ε_r_*]	4.0
Probability of crossover of real variable	0.9
Probability of mutation of real variable	0.5
Seed for random number generator	0.5
Number of crossovers of real variable	1796 for 100 generations
Number of mutations of real variable	3923 for 100 generations
Execution time	56 min 47 s for 100 generations

**Table 4 sensors-23-01685-t004:** Obtained values of *E_ac_* and *A*.

${\bar{U}}_{Test}\;(\mathrm{kV})$	3	4	5	6	7	8
***E*_ac_** (eV)	3.485	2.655	2.274	2.104	2.038	1.948
***A*** (µS/m)	5.440	0.391	0.117	0.008	0.006	0.004

## Data Availability

Not applicable.

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
