# Peer review of "Effective Electrical Properties and Fault Diagnosis of Insulating Oil Using the 2D Cell Method and NSGA-II Genetic Algorithm"

_sensors, 2023, doi:10.3390/s23031685_

Round 1
Reviewer 1 Report
the article is clear and well structured. The experimental setup is well described, and the research problem is clearly posed. The article does not present particularly innovative methods, but describes an application case well. There are, in my opinion, a few aspects concerning the GA model that need to be clarified.
The field in which the algorithm was used is that of optimisation, and is a case of Population-Based Evolutionary algorithms. The authors start from a set of possible solutions (the population), in order to generate new solutions that work better. It should be made clearer how the population is constituted, and also what features are associated with the chromosomes. What is the best representation of chromosomes for this problem? And what is the fitting? These aspects could also be included, to represent how the model is used.
Finally, the testing and validation methods should be dealt with a little more extensively.
Author Response
REVIEWER 1
The document has been checked and minor spelling mistakes have been corrected. Thank you very much for your detailed suggestions.
General Comments:
The article is clear and well structured. The experimental setup is well described, and the research problem is clearly posed. The article does not present particularly innovative methods, but describes an application case well. There are, in my opinion, a few aspects concerning the GA model that need to be clarified.
The field in which the algorithm was used is that of optimisation, and it is a case of Population-Based Evolutionary algorithms. The authors start from a set of possible solutions (the population), in order to generate new solutions that work better.
- It should be made clearer how the population is constituted, and also what features are associated with the chromosomes.
Answer:
To clarify this, we have added the following paragraphs in line 604:
“In step 1, an initial random population of size N equal to 40 individuals is generated. The execution time is approximately 56 minutes for 100 generations. In this problem, each individual of the population is equivalent to a chromosome which is constituted by the union of the physical properties to be determined, .
For the codification of the chromosome, the algorithm needs to know the minimum and maximum intervals of these variables. For , which is part of the chromosome, the values considered are a lower limit of 0.15 pS/m, which is a very low value, and an upper limit of 80 pS/m, which is a very high value. All the experiments carried out are between those two limits. In the same way, the lower and upper limit of are 1 and 4 respectively. 1 corresponds to the air and 4 is the maximum value found in the literature.”
- What is the best representation of chromosomes for this problem?
Answer:
As we say in the article “A real codification was chosen for each chromosome . In general, the precision is better than binary codification. It can be improved adding more bits, but this increases the simulation time [21]”.
As the variable of the parameter space of an optimization problem is continuous, a real coded GA is possibly preferred [41].
Therefore, in line 593 we have included:
“As the variable of the parameter space of an optimization problem is continuous, a real coded GA is possibly preferred [41]”.
[41] Gaffney, J.; Green, D. A.; Pearce, C. E. M. Binary versus real coding for genetic algorithms: A false dichotomy? Proceedings of the 9th Biennial Engineering Mathematics and Applications Conference, EMAC-2009. Vol. 51, pp. C347- C359. https://doi.org/10.21914/anziamj.v51i0.2776.
- And what is the fitting?
Answer:
To clarify this point, we have added the following paragraph in line 558:
“The objective functions used in the NSGA-II model define the error that is produced when we are searching for the final solutions of the iterative problem, and . In this way, we fitted the experimental data obtained in section 3 to the numerical solutions obtained through CM.”
These aspects could also be included, to represent how the model is used.
Finally, the testing and validation methods should be dealt with a little more extensively.
Answer:
In section 2.2, we obtained the capacity of the TDC using CM and the value obtained is equal to that provided by the manufacturer. This validates the CM numerical model that uses the geometry and the physical laws that govern the problem.
Therefore, we could apply any dielectric and any TDC geometry. The proposed method is a general method. It is not a specific method for a particular TDC.
In addition, in section 4.2, Figure 33, we validate the results obtained with the iterative method proposed in this article for with those obtained with the analytical expression of Arrhenius, and they are coincident for all the values that are a function of temperature and voltage.
To clarify this point, we have added in line 260 the following paragraph:
“As we obtained the capacity of the TDC using CM and the value obtained is equal to that provided by the manufacturer, we validated the CM numerical model that uses the geometry and the physical laws that govern the problem. Therefore, we could apply any dielectric and any TDC geometry. The proposed method is a general method. It is not a specific method for a particular TDC.”

Reviewer 2 Report
The paper presents the study results of effective electrical properties and fault diagnosis of insulating oil by the use of 2-D cell method and NSGA-II genetic algorithm. Authors made an analysis of the quality of electrical insulating oils. They measured electrical parameters, such as tan(delta). Authors proposed numerical method in order to determine the electrical resistivity, σ, and electrical permittivity, ε. Obtained results show improved accuracy compared to existing analytical equations.
Dear author, thank you very much for interesting paper about electrical resistivity and permittivity of oil using new, and promising methods. I put some comments and questions.
Comments:
1. Introduction chapter is well organized. Authors describe effect of higher temperature on transformer lifetime, where the low of 7C is presented. They discuss about electrical properties role which determines proper work of power transformer. Authors indicate the necessary to use an algorithm, which could help to operate transformer. They present evolution of various algorithms.
2. All devices and methods, described in chapter 2, are correct used by authors.
3. Fig.4. – Authors present a result of computer simulation of potential distribution. I think, not potential, only electric field stress would be more useful for investigations described in the paper. Please consider it.
4. line 240 – I am sure that range of tan(delta) values for oil is bigger, and the maximum value of the tan(delta) is bigger, especially for aged oil. Please correct it using very rich literature.
5. Fig.18 – presented results of tan(delta) as a function of used voltage and temperature are already well known. What was a reason to measure it? I did not find the answer.
6. Formula 9 describes relationships between tan(delta) and resistivity and permittivity. Why did authors measure tan(delta) if you have measured resistance and capacity, which could help to calculate tan(delta) using modified formula 9?
7. General conclusions: the paper is very interesting and value. I think, it is almost ready to be published but authors need to complete and explain some information.
Author Response
REVIEWER 2
The document has been checked and minor spelling mistakes have been corrected. Thank you very much for your detailed suggestions.
The paper presents the study results of effective electrical properties and fault diagnosis of insulating oil by the use of 2-D cell method and NSGA-II genetic algorithm. Authors made an analysis of the quality of electrical insulating oils. They measured electrical parameters, such as tan(delta). Authors proposed numerical method in order to determine the electrical resistivity, σ, and electrical permittivity, ε. Obtained results show improved accuracy compared to existing analytical equations.
Dear author, thank you very much for interesting paper about electrical resistivity and permittivity of oil using new, and promising methods. I put some comments and questions.
Comments:
- Introduction chapter is well organized. Authors describe effect of higher temperature on transformer lifetime, where the low of 7C is presented. They discuss about electrical properties role which determines proper work of power transformer. Authors indicate the necessary to use an algorithm, which could help to operate transformer. They present evolution of various algorithms.
- All devices and methods, described in chapter 2, are correct used by authors.
- Fig.4. – Authors present a result of computer simulation of potential distribution. I think, not potential, only electric field stress would be more useful for investigations described in the paper. Please consider it.
Answer:
Thank you very much. We agree with you. We have added a new figure with the electric field stress (Figure 4b: Distribution of the electric field strength modulus corresponding to the numerical experiment AG3 of section 4.1.). We think that this figure, together with Figure 4a which represents the voltage distribution, contributes to clarify the comprehension of the article.
We have added in line 240 the following text:
“The intensity of the electric field is the negative gradient of the electric potential, which is obtained by solving Eq.(2). The intensity of the electric field is used to calculate the stored energy and the heat losses as observed in Eqs. (4) and (5) which are necessary in the calculation of the effective parameters, σ and ε, as explained in section 4.1.
Figure 4b shows the distribution of the electric field intensity module corresponding to one of the optimization experiments carried out (AG3 in section 4.1.). The values of electrical conductivity and relative permittivity of this distribution are 20.7 pS/m and 2.11, respectively.”
- line 240 – I am sure that range of tan(delta) values for oil is bigger, and the maximum value of the tan(delta) is bigger, especially for aged oil. Please correct it using very rich literature.
Answer:
Thank you very much. You are right. We have looked for new references in the literature and found higher values, with bigger maximum values of the tan(delta), especially for aged oils. We have added references [33] and [34] which present respective values of 0.053 and 0.015, which are higher than the maximum value we gave for fresh oils. In addition, ASTM D924 - 15 [12] states that, in general, tangent delta values of insulation oils in good condition are below 0.005.
We have modified in the article line 240 “The value of for fresh oil can vary from 0.0010 to 0.0015 [29]”, and we have introduced a new paragraph in line 292:
“ASTM D924 - 15 [12] states that, in general, values of insulation oils in good condition are below 0.005. Some of the values found in the scientific literature are 0.001 [32], 0.053 [33] and 0.015 [34]. Higher values are found for aged oils.”
[33] Liao, R.; Feng, D.; Hao, J.; Yang, L.; Li, J. Thermal and Electrical Properties of a Novel 3-Element Mixed Insulation Oil for Power Transformers. IEEE Transactions on Dielectrics and Electrical Insulation 2019. Vol. 26, no. 2, pp. 610 - 617. https://doi.org/ 10.1109/TDEI.2019.007464.
[34] Sangineni, R.; Baruah, N.; Kumar Nayak, S. Study of the Dielectric Frequency Response of Mineral Oil on addition of Carboxylic Acids. IEEE Conference on Electrical Insulation and Dielectric Phenomena (CEIDP), Denver, CO, USA, 2022. pp. 143-146. https://doi.org/10.1109/CEIDP55452.2022.9985364.
- Fig.18 – presented results of tan(delta) as a function of used voltage and temperature are already well known. What was a reason to measure it? I did not find the answer.
Answer:
You are right. This value, tan(delta), is not used directly in the numerical model developed in section 4 to obtain the effective parameters, σ and ε. The paper follows the ASTM D 924-15 standard (standard test method for dissipation factor and relative permittivity of electrical insulating liquids), where effective electrical parameters, including the tan δ of the oil, are obtained. The parameters that are directly involved in the model are , parallel resistance of the equivalent circuit of lumped parameters, and , parallel capacity of the equivalent circuit of lumped parameters.
To clarify this, in line 561, we have added the following text:
“The paper follows the ASTM D 924-15 standard where effective electrical parameters, including the tan δ of the oil, are obtained. However, note that is not used directly in the numerical model developed in this section to obtain the effective parameters, σ and ε. The parameters that are directly involved in the model are , parallel resistance of the equivalent circuit of lumped parameters and , parallel capacity of the equivalent circuit of lumped parameters”.
- Formula 9 describes relationships between tan(delta) and resistivity and permittivity. Why did authors measure tan(delta) if you have measured resistance and capacity, which could help to calculate tan(delta) using modified formula 9?
Answer:
We do not use formula 9 in the calculations made in section 4 to obtain the effective parameters σ and ε because this formula is only an approximation which does not take into account the geometry of the TDC. However, the Cell Method uses it and, therefore, it is more precise.
- General conclusions: the paper is very interesting and value. I think, it is almost ready to be published but authors need to complete and explain some information.
